# Comprehensive Analysis of Autophagy-Related Genes in Rice Immunity against *Magnaporthe oryzae*

**DOI:** 10.3390/plants13070927

**Published:** 2024-03-22

**Authors:** Xuze Xie, Mengtian Pei, Shan Liu, Xinxiao Wang, Shanshan Gong, Jing Chen, Ye Zhang, Zonghua Wang, Guodong Lu, Ya Li

**Affiliations:** 1State Key Laboratory of Ecological Pest Control for Fujian and Taiwan Crops, Fujian Universities Key Laboratory for Plant-Microbe Interaction, College of Life Sciences, Fujian Agriculture and Forestry University, Fuzhou 350002, China; xxz2020phd@163.com (X.X.); 15870651077@163.com (M.P.); 18975260293@163.com (S.L.); 15806623113@163.com (X.W.); gss981030@163.com (S.G.); 15960063268@163.com (J.C.); 15059968641@163.com (Y.Z.); 2Fujian Provincial Quality Safety Inspection and Test Center for Agricultural Products, Fuzhou 350003, China; 3Fujian Key Laboratory on Conservation and Sustainable Utilization of Marine Biodiversity, Fuzhou Institute of Oceanography, Minjiang University, Fuzhou 350108, China; 4Ministerial and Provincial Joint Innovation Centre for Safety Production of Cross-Strait Crops, Minjiang University, Fuzhou 350108, China; 5Fujian Key Laboratory for Monitoring and Integrated Management of Crop Pests, Fuzhou 350013, China

**Keywords:** autophagy, OsATGs, plant immunity, salicylic acid, jasmonic acid, *Magnaporthe oryzae*

## Abstract

Rice blast disease, caused by the fungus *Magnaporthe oryzae*, is a significant threat to rice production. Resistant cultivars can effectively resist the invasion of *M. oryzae*. Thus, the identification of disease-resistant genes is of utmost importance for improving rice production. Autophagy, a cellular process that recycles damaged components, plays a vital role in plant growth, development, senescence, stress response, and immunity. To understand the involvement of autophagy-related genes (ATGs) in rice immune response against *M. oryzae*, we conducted a comprehensive analysis of 37 OsATGs, including bioinformatic analysis, transcriptome analysis, disease resistance analysis, and protein interaction analysis. Bioinformatic analysis revealed that the promoter regions of 33 OsATGs contained cis-acting elements responsive to salicylic acid (SA) or jasmonic acid (JA), two key hormones involved in plant defense responses. Transcriptome data showed that 21 OsATGs were upregulated during *M. oryzae* infection. Loss-of-function experiments demonstrated that OsATG6c, OsATG8a, OsATG9b, and OsATG13a contribute to rice blast resistance. Additionally, through protein interaction analysis, we identified five proteins that may interact with OsATG13a and potentially contribute to plant immunity. Our study highlights the important role of autophagy in rice immunity and suggests that OsATGs may enhance resistance to rice blast fungus through the involvement of SA, JA, or immune-related proteins. These findings provide valuable insights for future efforts in improving rice production through the identification and utilization of autophagy-related genes.

## 1. Introduction

Autophagy is a cellular recycling mechanism that involves the degradation and recycling of damaged or unnecessary cellular components, such as proteins, organelles, and pathogens, to maintain cellular homeostasis and promote cell survival [1,2]. Based on the different degradation targets and degradation mechanisms, autophagy can be divided into three types: macroautophagy, microautophagy, and selective autophagy [1]. The most commonly referenced type is macroautophagy, wherein the substrates to be degraded are enveloped by structures called autophagosomes and transported to lysosomes or vacuoles for degradation [3]. The process of autophagy can be further subdivided into several stages, namely induction, nucleation, elongation, closure, transport, and degradation [4].

Autophagy-related genes (ATGs) are a group of genes involved in the autophagy process. They are responsible for functions such as forming the autophagosome membrane, recruiting cargo to the autophagosome, and facilitating the fusion of autophagosomes with lysosomes for degradation [5]. These ATGs are highly conserved across different species, including plants, animals, and fungi. The first ATG was identified in *Saccharomyces cerevisiae*, and since then, over 40 ATGs have been identified in eukaryotic organisms [6,7]. In various plant model organisms, such as *Arabidopsis thaliana*, *Zea mays*, *Oryza sativa*, *Rosa roxburghii*, and *Camellia sinensis*, ATGs have been extensively studied and found to be involved in growth, development, senescence, stress response, and immunity [8,9,10,11].

Autophagy plays a crucial role in the growth and development of plants. ATGs have been identified as key players in the aging process in various plant species, including *Arabidopsis thaliana*, *Zea mays*, *Hordeum vulgare*, *Glycine max*, *Petunia hybrida*, and *Brassica rapa* [8,12,13,14,15,16]. In *Arabidopsis*, ATGs are known to regulate the production of reactive oxygen species (ROS) and indole-3-acetic acid (IAA), thus influencing glucose-mediated root meristem growth [17]. Additionally, the knockout of OsATG7 in rice has been shown to have a significant impact on seed size and starch content, highlighting the essential role of autophagy in seed development [18].

Autophagy works as a fundamental defense response and stress regulatory mechanism that promotes cell repair and regeneration. When plants are subjected to biotic and abiotic stresses, autophagy is significantly activated to facilitate organismal repair [19]. Defects in autophagy make plants more susceptible to abiotic stress [20]. For instance, the silencing of *Arabidopsis* AtATG18 leads to increased drought sensitivity [21]. On the other hand, the overexpression of MdATG18 in *Malus domestica* enhances apple tolerance to drought stress [22]. Under cold stress, the expression of rice *OsATG6a/c* is downregulated [23], while barley *HvATG6* is upregulated [24]. Moreover, the functional deficiency of wheat TaATG7 and rice OsATG10b reduces plant tolerance to salt stress [25,26], whereas the overexpression of MdATG10 in apples increases their tolerance to salt stress [27].

Autophagy plays a crucial role in plant immunity and defense against pathogen invasion [28]. For example, AtATG18a interacts with AtWRKY33 to aid plants in defending against necrotrophic pathogens [29]. In *Solanum lycopersicum*, the silencing of SlATG5, SlATG7, and SlATG8h promotes infection by TLCYnV (tomato leaf curl Yunnan virus) [30]. Similarly, the loss of function of OsATG5 or OsATG7 in rice leads to increased susceptibility to RSV (rice stripe virus), highlighting the role of autophagy in suppressing RSV infection [31].

Rice blast, caused by the fungus *Magnaporthe oryzae*, is a severe disease that greatly impacts rice production and global yield. However, the understanding of how rice autophagy, an essential plant immunity regulatory mechanism, contributes to rice resistance against blast disease remains limited. In 2011, Xia et al. discovered 33 ATGs in rice, marking an early milestone in autophagy research [8]. Since then, with advancements in autophagy research and the refinement of rice gene annotation, more OsATGs (ATGs in *O. sativa*) have been identified. Building upon the foundation laid by Xia et al., we further refined the list by excluding five OsATGs lacking the ATG domain or gene ID. Moreover, we included nine OsATGs that exhibited ATG-related structural domains. The primary objective of this study is to investigate the role of 37 OsATGs in rice immunity against blast disease.

## 2. Results

### 2.1. Identification of 37 OsATGs in the Rice Genome

In this study, we identified a total of 37 ATGs in the rice genome (Appendix A), designated as OsATGs (*Oryza sativa* ATGs). Among these 37 OsATGs, 28 were in the list of 33 OsATGs reported by Xia K et al [8]. Five OsATGs in Xia’s list were excluded in our study, because three of them, OsATG1b (Os03g0289100), OsATG1c (Os08g0484600), and OsATG16 (Os09g0497700), were no longer associated with autophagy-related proteins; and two of them, OsATG8f and OsATG8i, lacked official gene numbers (RAP-DB or MSU Locus ID). Additionally, we included nine new OsATGs to our study based on the presence of autophagy-related domains: OsATG1b (Os04g0686600), OsATG1c (Os03g0268200), OsATG2 (Os06g0267600), OsATG11 (Os02g0179800), OsATG13c (Os11g0162000), OsATG14a (Os06g0715000), OsATG14b (Os07g0626300), OsATG16 (Os03g0746600), and OsATG18g (LOC_Os07g47410). It is noteworthy that OsATG1c is the homolog of both AtATG1b and AtATG1c, OsATG1b is the homolog of AtATG1d, and OsATG13c is the homolog of AtATG13b. The remaining six newly added OsATGs correspond to the homologs of AtATGs in *Arabidopsis thaliana*.

### 2.2. Physicochemical Properties and Subcellular Localization of OsATGs

As shown in Table 1, different OsATG proteins exhibit varying physicochemical properties. The amino acid count ranges from 93 to 1919, while the molecular weights range from 10,461.01 to 212,230.11 Da. The theoretical pI (isoelectric point) ranges from 4.52 (OsATG3a) to 9.71 (OsATG14b), with an average of 6.81. Out of the 37 OsATG proteins, 12 are considered alkaline (pI > 7.5), while 25 are considered acidic (pI < 7.5). Furthermore, thirty OsATG proteins are stable (with an instability index exceeding 40, which is the threshold for stable proteins), while the remaining seven are unstable. The lipid coefficient (shown as the aliphatic index) ranges from 65 to 102.36. Hydrophobicity analysis showed that thirty-six OsATG proteins are hydrophilic (GRAVY < 0), while only one is hydrophobic (GRAVY > 0). The prediction of protein subcellular localization showed that these OsATG proteins primarily reside in the nucleus, inner membrane, and organelle membranes. Additionally, we performed tertiary structure analyses of these OsATG proteins and found that different homologous proteins of the same OsATG have similar protein conformations (Appendix A).

### 2.3. Phylogenetic Relationship and Chromosomal Localization of OsATGs

To investigate the phylogenetic relationship of OsATG proteins, we performed cluster analysis using the neighbor-joining method in MEGA11.0 software. The results revealed that OsATG proteins can be divided into five categories (Figure 1a). Eight OsATG proteins, including OsATG2, OsATG5, OsATG7, OsATG11, OsATG12, OsATG14a, OsATG14b, and OsATG16, belong to distinct evolutionary branches, indicating their independent evolution. The remaining 29 OsATG proteins can be further divided into groups I, II, III, and IV, with variations in the number of proteins in each group. Group I consists solely of three OsATG13 proteins; group II consists solely of seven OsATG18 proteins; group III includes two OsATG3 proteins, two OsATG4 proteins, three OsATG6 proteins, and two OsATG10 proteins; the remaining ten OsATG proteins are clustered in group IV. These findings provide insights into the evolutionary relationships and diversification of OsATG proteins.

Furthermore, we analyzed the chromosomal localization of OsATGs. The results showed that the 37 OsATGs are distributed unevenly across 11 chromosomes, excluding Chr 9 (Figure 1b). Chr 3 contains the highest number of genes (seven), followed by Chr 1 (six), while Chr 8 and Chr 12 have the fewest genes (one each). Interestingly, we found that approximately one-third of the genes are located near the telomeric regions of the chromosomes, while the majority of the remaining genes are situated in the distal regions away from the telomeres. These findings provide valuable insights into the spatial distribution patterns of OsATGs on the rice chromosomes.

### 2.4. Gene Structure and Function Domain of OsATGs

Analysis of the gene structure of OsATGs revealed variations in the number of exons among the members (Figure 2a). Notably, *OsATG6c* has the fewest with one exon, while *OsATG7* has the most with twenty exons. The remaining genes have varying numbers of exons ranging from two to thirteen. This uneven distribution of exons suggests potential intron insertion or deletion events during the evolution of *OsATGs*. Further examination of the functional domains of OsATGs unveiled that all members harbor typical autophagy-related domains, as illustrated in Figure 2b. Specifically, three OsATG1s feature a kinase domain, while one OsATG2, two OsATG3s, and two OsATG10s contain an ATG domain at the C-terminal region. Additionally, two OsATG4s possess a Peptidase-C54 domain, and all seven OsATG18s include a WD40 domain. The remaining OsATGs each contain a specific autophagy-related domain. These findings suggest that OsATGs are involved in a diverse array of biological processes.

### 2.5. Protein Motif of OsATGs

Motif prediction revealed that, among 37 OsATG proteins, 26 contain motifs, with a total of 20 motifs identified (Figure 3). These motifs vary in terms of number, type, and length. The shortest motif (motif 12) is only 21 amino acids long and is present in five OsATG8 proteins. On the other hand, the longest motifs (motifs 1–5, 7, 8, 14, 19, and 20) span 50 amino acids and are found in ten OsATG proteins. All 26 OsATG proteins contain these longest motifs, except for OsATG13a, OsATG13b, and OsATG16. Interestingly, it was observed that three OsATG1 proteins and three OsATG6 proteins contain motif 5. Additionally, motif 1 is present in five OsATG8 proteins, as well as OsATG9a and OsATG11. motif 11 is found exclusively in OsATG13a and OsATG13b. Furthermore, all five OsATG18 proteins contain motif 3. Notably, OsATG6a, OsATG6b and OsATG6c have the highest number of motifs, suggesting their involvement in multiple pathways. These findings indicate that OsATGs can be regulated though various motifs, allowing them to serve diverse functions within the organism.

### 2.6. Genetic cis-Regulatory Elements of OsATGs

The analysis of cis-regulatory elements (CREs) was performed on the 2 kb upstream promoter sequence of OsATGs. The results revealed that there are 24 types of CREs in the promoter region (Figure 4a). The study primarily focused on the involvement of these CREs in hormone regulation and stress response. The CREs of hormone regulation include response to salicylic acid (SA), methyl jasmonate (MeJA), ethylene (ETH), abscisic acid (ABA), gibberellin (GA), and auxin (AUX). And the CREs of stress response include responses to low temperature, drought, wounding, and defense. Analyzing the hormone regulatory CREs, we found that the promoter region of 13 *OsATG* genes contain CREs of SA response, and 30 *OsATGs* contain MeJA, 16 *OsATGs* contain ETH, 34 *OsATGs* contain ABA, 20 *OsATGs* contain GA, 17 *OsATGs* contain AUX (Figure 4b). This significant enrichment of hormone CREs suggests that OsATGs play crucial roles in various biological processes mediated by different plant hormones.

### 2.7. Resistance to Rice Blast Fungus of OsATGs

Using the available transcriptome data during infection by *M. oryzae* in rice, a hierarchical clustering heatmap was generated to analyze the expression patterns of *OsATGs*. Out of the 37 *OsATG* genes analyzed, with the exception of *OsATG8e* and OsATG18g, all were expressed and showed differential expression patterns (Figure 5a and Appendix A). The results also showed that 17 *OsATGs* exhibited increased expression at 24 h post-infection, while 10 *OsATGs* showed increased expression at 48 h. Notably, six *OsATGs* (*OsATG1b*, *OsATG6a*, *OsATG6c*, *OsATG10a*, *OsATG12*, and *OsATG13b*) displayed sustained increased expression throughout the 0–48 h period. These results strongly suggest that autophagy plays a significant role in conferring resistance to rice blast fungus.

To further investigate the involvement of OsATGs in rice immunity against blast fungus, we conducted a disease resistance analysis on six loss-of-function mutants of OsATGs (OsATG6b, OsATG6c, OsATG8a, OsATG9a, OsATG9b, and OsATG13a). The results of inoculating rice with *M. oryzae* showed that the relative lesion area of the *osatg8a* and *osatg13a* mutants significantly increased compared to the wild type, while there was no significant change in the other mutants (Figure 5b). Interestingly, when wounded rice was inoculated, the *osatg6c* and *osatg9b* mutants also showed enhanced susceptibility to *M. oryzae* (Figure 5c). These results indicate that OsATG6c, OsATG8a, OsATG9b, and OsATG13a play a positive role in regulating resistance to rice blast. The qRT-PCR results showed that OsATG6b and OsATG9a were significantly upregulated at various time points during *M. oryzae* infection (Figure 5d), suggesting that they are also involved in responding to *M. oryzae* infection. Taken together, these findings provide compelling evidence that rice OsATGs play a crucial role in immunity against blast fungus.

### 2.8. Response to Hormone Treatments of OsATGs

In order to investigate whether OsATGs are involved in the SA and JA signaling pathways, we analyzed the expression levels of *OsATG6b*, *OsATG6c*, *OsATG8a*, *OsATG9a*, *OsATG9b*, and *OsATG13a* genes in the wild-type ZH11 after induction with SA and MeJA at 0, 6, 12, and 36 h. The results showed that five *OsATG* genes (*OsATG6b*, *OsATG6c*, *OsATG8a*, *OsATG9b*, and *OsATG13a*) were responsive to the SA signaling pathway (Figure 6a), while five *OsATG* genes (*OsATG6c*, *OsATG8a*, *OsATG9a*, *OsATG9b*, and *OsATG13a*) were responsive to the SA signaling pathway (Figure 6b). Under SA induction, *OsATG6b* and *OsATG8a* were rapidly upregulated at 6 h, indicating their potential involvement in the rapid response to the SA pathway. *OsATG9b* and *OsATG13a* were upregulated from 12 to 36 h, while the expression level of *OsATG6c* decreased at 12 h. Similarly, under JA induction, the expression levels of four genes (*OsATG6c*, *OsATG8a*, *OsATG9b*, and *OsATG13a*) significantly increased at 6 h, demonstrating a rapid response to the JA signal, while the expression level of *OsATG9a* decreased at 36 h. Based on the above data, we hypothesize that OsATGs may participate in rice growth, development, and stress response mediated by SA or JA signaling pathways.

### 2.9. Subcellular Localization and Role of Immune Reaction of OsATG13a

Based on our analysis of pathogenicity and gene expression patterns induced by the plant hormones SA and JA, we have discovered that OsATG13a plays an important role in rice resistance against the blast fungus. To further investigate its function, we conducted a subcellular localization assay on OsATG13a. The results showed that OsATG13a was localized in the cytoplasm, with a strong punctate localization (Figure 7a). This punctate localization is consistent with the positioning of ATG16 in the phagosome [32]. Furthermore, our investigation revealed that GFP-OsATG13a co-localized with RFP-ATG16, confirming that the strong punctate localization of OsATG13a corresponds to the phagosome (Figure 7b).

Additionally, we examined the effects of chitin treatment on callose accumulation and the burst of ROS in the osatg13a mutant. Our results demonstrate that the osatg13a mutant exhibits significantly lower levels of both callose accumulation and ROS burst compared to the wild type under chitin induction (Figure 7c,d). These findings suggest that the loss function of OsATG13a leads to a reduced PTI response in rice against the blast fungus.

### 2.10. Possible Interaction Proteins of OsATG13a

We conducted a yeast two-hybrid screen to identify potential interacting proteins of OsATG13a. From this screen, we obtained a total of 104 transformants. After sequencing and verification, we identified 50 proteins that may interact with OsATG13a (Figure 8a and Appendix A). To further understand the roles of these proteins, we conducted a Gene Ontology (GO) analysis and found that they are involved in catalytic activity, cellular metabolism, and structural formation (Figure 8b). Notably, among these proteins, we discovered that five, namely OsRLK170576, OsKEG1, OsANR1, OsPP2A-4, OsVPE1, and OsMDH1, are potentially associated with plant immune responses (Table 2). This suggests that OsATG13a may play a role in the immune processes of rice through its interaction with these proteins.

## 3. Materials and Methods

### 3.1. Identification of OsATGs in Rice

A BLAST search was conducted in NCBI to identify ATG homologs in rice, with *Arabidopsis* AtATGs as the reference. It is to be noted that the search was specifically confined to the *Oryza sativa* Japonica Group (taxid: 39,947). A second round of searching was performed using the identified rice ATG homologs to ensure comprehensive coverage, while maintaining focus on the *Oryza sativa* Japonica Group (taxid: 39,947). The ATG gene IDs in three databases and the functional domains in NCBI were also searched. Detailed information on OsATGs is provided as Appendix A.

### 3.2. Bioinformatics Characterization Analysis

The mRNA and chromosome positions of OsATGs were obtained from the NCBI website. The Protparam website was used to calculate the physicochemical properties of OsATGs [39]. The Busca website was employed to predict the subcellular localization of OsATGs [40]. The functional domains of OsATGs were obtained from the SMART website [41]. The MEME website was used to identify conserved motifs of OsATGs [42]. The PlantCARE website was utilized to identify cis-acting elements in the upstream 2 kb promoter regions of OsATGs [43]. Data visualization was performed using the TBtools v1.098 software. The MEGA v11.0 software was used to construct a phylogenetic tree using the neighbor-joining method [44], and the tree was subsequently modified and refined using the Evolview website [45]. The SWISS-MODEL website was employed to predict the 3D structure of the proteins encoded by OsATGs [46]. The online URLs for all websites can be found in Appendix A.

### 3.3. Analysis of Gene Expression Patterns

Rice leaves were collected at 0 h, 24 h, and 48 h after infection with the rice blast fungus. The collected leaves were sent to Igenebook (Wuhan igenebook Biotechnology Co., Ltd., Wuhan, China) for RNA-seq analysis. The company performed expression analysis of various genes in rice. Subsequently, the expression levels of OsATG genes in rice were obtained and visualized using TBtools software.

### 3.4. Plants and Materials

All experiments were performed using *Oryza sativa L. ssp. Japonica cv. Zhonghua 11*. Mutants of rice were generously provided by Prof. Qingjun Xie (South China Agricultural University, Guangzhou, China). The specific mutation sites of each mutant can be found in Appendix A. Unless otherwise specified, the rice plants were cultivated under the following conditions: a temperature of 26 °C, relative humidity of 50%, light intensity of 200 μmol m^−2^s^−1^, and a 12 h photoperiod. Plasmids pHF223 and pHF225 were utilized for the construction of GFP or RFP fusion proteins, and the primer sequences can be found in Appendix A. For pathogen treatment, a concentration of 1–50 × 10^5^ mL^−1^ of rice blast fungus conidial suspension was employed. Regarding hormone treatment, SA and MeJA were added at concentrations of 1 mmol and 0.1 mmol, respectively, and the solvents used were sterile water containing 0.01% (*v*/*v*) Tween 20.

### 3.5. Disease Resistance Analysis

For spray inoculation, a 5 mL *M. oryzae* conidial suspension, containing 1 × 10^4^ mL^−1^, was evenly sprayed onto the 2-week-old rice seedlings using a sprayer pump bottle. For wound inoculation, the leaves of 8-week-old rice plants were gently punctured and then injected with 0.007 mL conidial suspension. Post inoculation, the rice plants were sealed for 24 h [47], and then were cultivated at a temperature of 26 °C and a relative humidity of 90% for a period of 5–7 days. The disease level of the rice plants was evaluated and captured through photodocumentation. The relative area of the lesions was calculated using ImageJ v1.46 software.

### 3.6. Pathogen Infection and Hormone Induction Treatment

SA, MeJA, or a conidial suspension was applied through spraying onto 2-week-old rice plants to investigate the expression of ATG genes at different time intervals. Total RNA was extracted from rice leaves using the Eastep^®^ Super Total RNA Extraction Kit (LS1040, Promega, Fitchburg, MA, USA). Subsequently, the RNA was reverse transcribed into cDNA using the 1st Strand cDNA Synthesis SuperMix for qPCR (11141ES60, Yeasen Biotechnology, Shanghai, China). The qRT-PCR analysis of rice OsATG genes was conducted using the qPCR SYBR Green Master Mix (11201ES08, Yeasen Biotechnology, Shanghai, China). The primer sequences can be found in Appendix A.

### 3.7. Subcellular Localization and Immune Response Analysis

The GFP (green fluorescent protein)—ATG and RFP (red fluorescent protein)—ATG fusion plasmids were constructed and then transformed into rice protoplasts. Following a transformation incubation lasting 12–24 h, the subcellular localization of ATG proteins was visualized using microscopy. Immune response experiments were conducted to evaluate callose deposition and reactive oxygen species (ROS) burst. These experiments were carried out following the methods described in a previously published article [48].

### 3.8. Analysis of Protein Interactions

The yeast two-hybrid screen was conducted to screen potential interacting proteins of the target protein. The BD-ATG plasmid was transformed with the plasmids of the AD-rice cDNA library into the yeast strain AH109. Subsequently, the resulting yeast transformants were subjected to amplification utilizing SF and SR primers, following which the amplified fragments were sent for sequencing. The obtained sequencing results were then aligned with sequences accessible on the NCBI database. Finally, the identified proteins underwent Gene Ontology (GO) analysis via the PANTHER website.

## 4. Discussion

*M. oryzae* is a pathogen that can infect various cereals [49]. The economic impact of this fungus is significant, with annual losses estimated at around $66 billion, and the destruction of harvested grain could have fed 60 million people [50]. While farmers can control rice blast through measures like crop rotation, proper water management, and judicious fertilizer use, breeding resistant cultivars is also effective. However, no cultivar is completely resistant to all races of *M. oryzae*, and resistant cultivars often lose their effectiveness within a few years as the fungus adapts [49,50]. Therefore, it is crucial to identify candidate disease-resistant genes and understand the underlying mechanisms of immune defense responses to increase rice yield.

Unlike animals, plants must overcome or tolerate various environmental stresses such as high temperatures, freezing, drought, flooding, herbivory by animals and insects, and invasion by pathogens [51,52]. Autophagy, a process that helps maintain cellular homeostasis, plays a significant role in helping plants cope with these stresses [53,54]. Plant ATG genes have been found to be involved in pathogen infection responses. However, the roles of different plant ATGs in plant immunity can vary due to variations in nutritional types and evolutionary directions among pathogens, and in some cases, they may even have opposite effects [55]. To investigate the role of rice OsATGs in response to *M. oryzae* infection, we conducted bioinformatic analysis, gene expression pattern analysis, and disease resistance analysis on OsATGs. Our study aims to provide a theoretical foundation for understanding the role of autophagy in plant immune processes.

In this study, we identified 37 OsATGs in rice and conducted a series of bioinformatic analyses. Our analyses revealed significant variations in the physicochemical properties of different OsATGs (Table 1) and variations in the number of exons (Figure 2a). However, the distribution of functional domains and conserved motifs exhibited similarities. For example, all OsATG8 genes contained the “ubl-ATG8” domain and motif 1, while all OsATG18 genes contained the “WD40” domain and motif 3 (Figure 2b and Figure 3a). These analytical results show similarities to the analysis of ATG homologs in other species [11,30].

Plant hormones play a crucial role in the growth and development of plants. However, they also function as cell signaling molecules and play an important role in defending against pathogen invasion [56]. Salicylic acid (SA) and jasmonic acid (JA) are key participants in plant immunity. They are able to enhance immune signal transduction and activate the expression of defense-related genes, thus promoting disease resistance [57,58,59]. In this study, we found that 13 and 26 promoter regions of OsATGs contain cis-acting elements responsive to SA and JA, respectively. Among them, six OsATGs were found to respond to both SA and JA simultaneously (Figure 4b). In the subsequent SA and JA induction experiments, the results showed that five genes responded to SA and JA induction, and among them, four genes (*OsATG6c*, *OsATG8a*, *OsATG9b*, and *OsATG13a*) were able to simultaneously respond to SA and JA induction (Figure 6). These results suggest the involvement of rice OsATGs in plant immune responses mediated by SA and JA.

Studying the gene expression patterns of plants during the infection stage of pathogens is crucial for identifying important candidate resistance genes [60]. Additionally, it allows for the rapid detection of the expression status of genes involved in plant immune processes [60]. Investigating the gene expression changes of plant ATG genes during the infection stage can provide insights into the role of autophagy in plant immunity. For instance, studies on banana and cassava have shown that there are eight and twenty-five upregulated ATG genes, respectively, during pathogen infection [61,62]. In our study, we observed diverse changes in the expression of *OsATGs* after 0, 24, and 48 h of *M. oryzae* infection. Specifically, seven *OsATGs* consistently showed upregulation, while ten *OsATGs* consistently showed downregulation (Figure 5a). During the infection stage of rice blast fungus, we observed significant changes in the gene expression of five *OsATGs* compared to 0 h (Figure 5d). These findings suggest that different *OsATGs* may play distinct roles in response to *M. oryzae* infection.

Autophagy genes can be categorized into four systems, including the ATG1-ATG13 complex, the ATG9-ATG2 complex, the PI3K-ATG6 complex, and the ATG8-ATG12 ubiquitination system. Each system has a distinct function in autophagy, including autophagy initiation, membrane supply for autophagosome formation, nucleation of autophagosomes, and membrane expansion of autophagosomes, respectively [1,63]. To comprehensively elucidate the role of autophagy genes in rice immunity, we focused on studying one to two genes from each system, specifically OsATG13a, OsATG9a, OsATG9b, OsATG6b, OsATG6c, and OsATG8a. Resistance assays revealed that loss of function in OsATG6c, OsATG8a, OsATG9b, and OsATG13a led to decreased resistance against *M. oryzae* (Figure 5b,c). Furthermore, the *osatg13a* mutant showed reduced chitin-induced callose accumulation and ROS burst compared to the wild type (Figure 7). These findings suggest that autophagy in rice positively regulates the immune response to *M. oryzae*.

In this study, we have identified several key autophagy-related genes (OsATG6c, OsATG9b, OsATG8a, and OsATG13a) that contribute to rice resistance against *M. oryzae* infection. Interestingly, the role of ATG6 in plant immunity can vary depending on the specific pathogen. For instance, the loss of function of BrATG6 in cabbage (*Brassica rapa*) leads to increased susceptibility to *Plasmodiophora brassicae*, while the absence of TaATG6 in wheat enhances resistance against *Blumeria graminis* f. sp. *tritici* [64,65]. Moreover, ATG9 and ATG8 have been found to play a positive regulatory role in plant immunity. The overexpression of NbATG9 in tobacco (*Nicotiana benthamiana*) significantly enhances autophagy activity, resulting in increased resistance against *Phytophthora infestans*, the causal agent of potato late blight [66]. Similarly, in wheat, TaATG8j enhances resistance against *Puccinia striiformis* f.sp. *tritici*, the causal agent of stripe rust, by regulating cell death [67]. However, there is limited information on the involvement of ATG13 in plant immunity. Therefore, in order to gain a better understanding of the potential mechanisms underlying its involvement in rice immunity, we conducted a screening for interacting proteins of OsATG13a. Our preliminary results suggest that OsRLK170579, OsKEG1, OsANR1, OsPP2A-4, OsVPE1, and OsMDH1 may be involved in plant immunity by interacting with OsATG13a.

Receptor-like kinases (RLKs) are essential protein kinases that are widely present in plants and play essential roles in growth, development, and immunity. Yin and Liu have developed RLKdb, a comprehensive RLKs database containing approximately 220,000 RLKs from 300 plant genomes [33]. Among the RLKs included in the database, OsRLK170576 stands out as a potential key player in rice immunity. The E3 ligase KEEP ON GOING (KEG) is involved in ubiquitination processes. In *Arabidopsis*, KEG regulates the protein levels of MKK4 and MKK5 through ubiquitination, revealing a mechanism through which plants can finely regulate immune responses by modulating key MAPK cascade members [34]. Proanthocyanidins (PAs), also known as condensed tannins, are significant defense-related phenolic compounds in mulberry trees (*Morus alba*), and their biosynthesis is linked to the expression of anthocyanidin reductase (ANR). Studies have shown that the heterologous expression of MnANR from mulberry in tobacco enhances tobacco’s resistance to *Botrytis cinerea* [35]. Serine/threonine protein phosphatases type 2A (PP2A) play crucial roles in growth, development, stress responses, and hormone signaling. In the potato (*Solanum tuberosum*), the overexpression of StPP2Ac2b, a member of PP2A, leads to plant senescence, resulting in decreased resistance to late blight disease [36]. Vacuolar processing enzyme (VPE) is a cysteine protease involved in the maturation of vacuolar protein. In *N. benthamiana*, NbVPE plays a role in the hypersensitive response (HR) induced by elicitors, suggesting its involvement in immune processes triggered by elicitors [37]. Malate dehydrogenase (MDH) is a critical metabolic enzyme involved in various plant development processes. In cassava (*Manihot esculenta*), MeMDH1 positively regulates cassava’s resistance to *Xanthomonas manihotis* by modulating the accumulation of SA and the expression of pathogenesis-related protein 1 [38]. Therefore, it is plausible that rice OsATG13a may participate in rice immune processes through interactions with OsRLK170576, OsKEG1, OsANR1, OsPP2A-4, OsVPE1, and OsMDH1.

In this study, we conducted a comprehensive analysis of 37 homologs of autophagy-related (ATG) proteins in rice. Our analysis provided valuable insights into the bioinformatic features of rice autophagy genes, their hormone-induced responses, and their involvement in rice disease resistance. By analyzing the gene expression patterns of OsATGs, we observed differential gene expressions in response to infection by the rice blast fungus, which can potentially aid in the identification of candidate disease resistance genes. Furthermore, through the analysis of disease resistance in mutants lacking certain OsATGs, we identified the significant roles of OsATG6c, OsATG9b, OsATG8a, and OsATG13a in rice resistance, enhancing our understanding of the relationship between autophagy and rice immunity. Additionally, we conducted an analysis of the probable interacting proteins of OsATG13a, leading to the identification of six genes that potentially interact with OsATG13a and may participate in plant immunity. Overall, our study provides valuable references for rice breeding and safe production in terms of disease resistance. It contributes to the development of new resistant cultivars and enhances the overall disease resistance capacity of rice. Future research should focus on elucidating the relationship between rice autophagy and rice immunity, studying the impact of altered autophagy activity on rice disease resistance, and investigating potential signal transduction between rice OsATGs and RLKs.

## 5. Conclusions

In this study, our findings demonstrate the crucial role of rice autophagy in the immune system. And rice OsATGs may participate in the immune response against rice blast fungus by the induction of SA or JA, and the interaction of immune-related proteins. These significant findings provide a solid theoretical foundation for further exploration of the role of plant autophagy in plant immunity.

## Figures and Tables

**Figure 1 plants-13-00927-f001:**
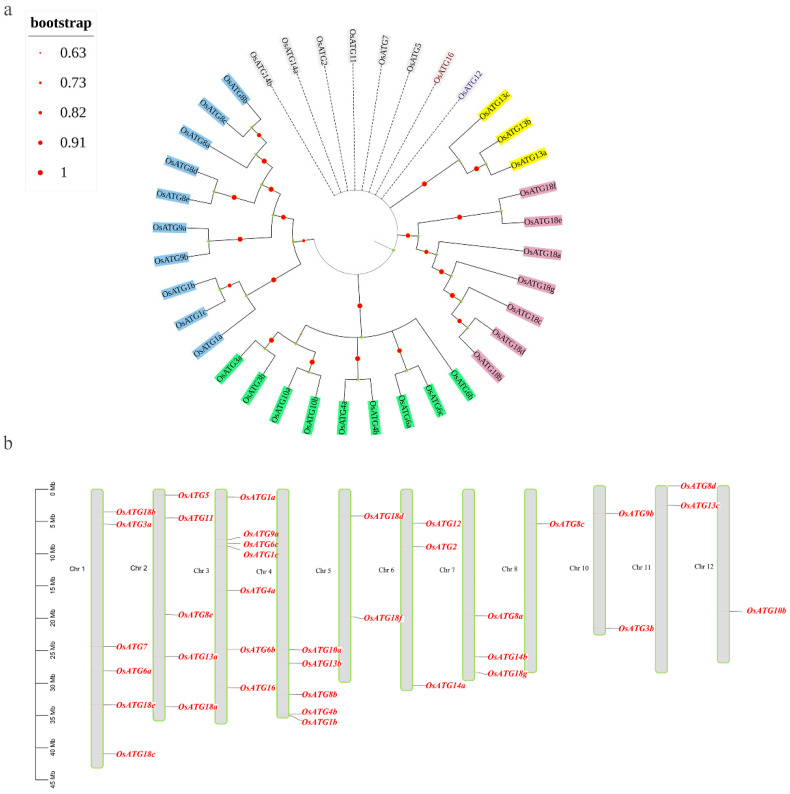
Phylogenetic and chromosomal localization analysis of OsATGs. (**a**) Phylogenetic analysis of OsATGs. The bootstrap values indicate the confidence of the branching. Clusters I–IV are represented by yellow, purple, green, and blue colors, respectively, while gray indicates individual groups. (**b**) Chromosomal localization analysis of OsATGs. The scale represents the distance on the chromosome, and the number of chromosomes is shown on the left side of each chromosome. Chromosomes are represented by gray boxes with green borders, while *OsATG* genes are displayed in red font. Rice has a total of 12 chromosomes, and *OsATG* genes are not distributed on Chr 9; hence, it is not shown here.

**Figure 2 plants-13-00927-f002:**
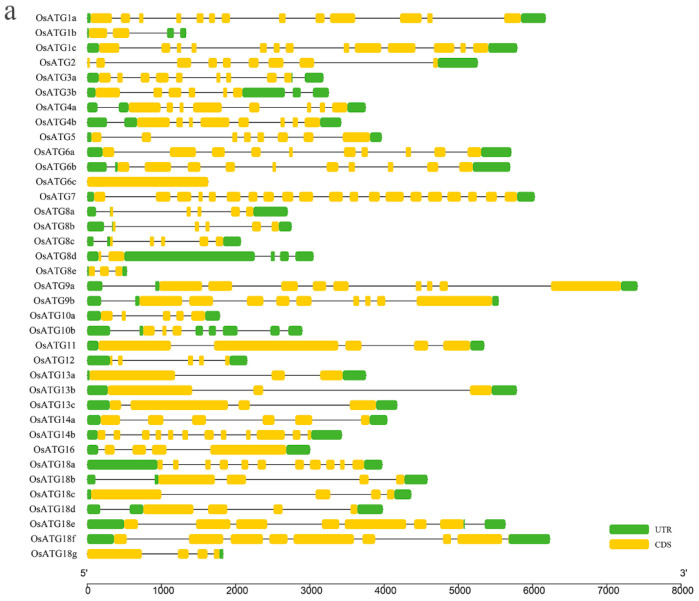
Gene structure and functional domain analysis of OsATGs. (**a**) Gene structure analysis of *OsATG* genes. The scale represents the gene length of OsATGs. The green box represents the untranslated region (UTR), the yellow box represents the coding sequence (CDS), and the unmarked regions represent introns. (**b**) Functional domain analysis of OsATG proteins. The scale represents the protein size of OsATGs, and the different colored boxes represent different functional domains.

**Figure 3 plants-13-00927-f003:**
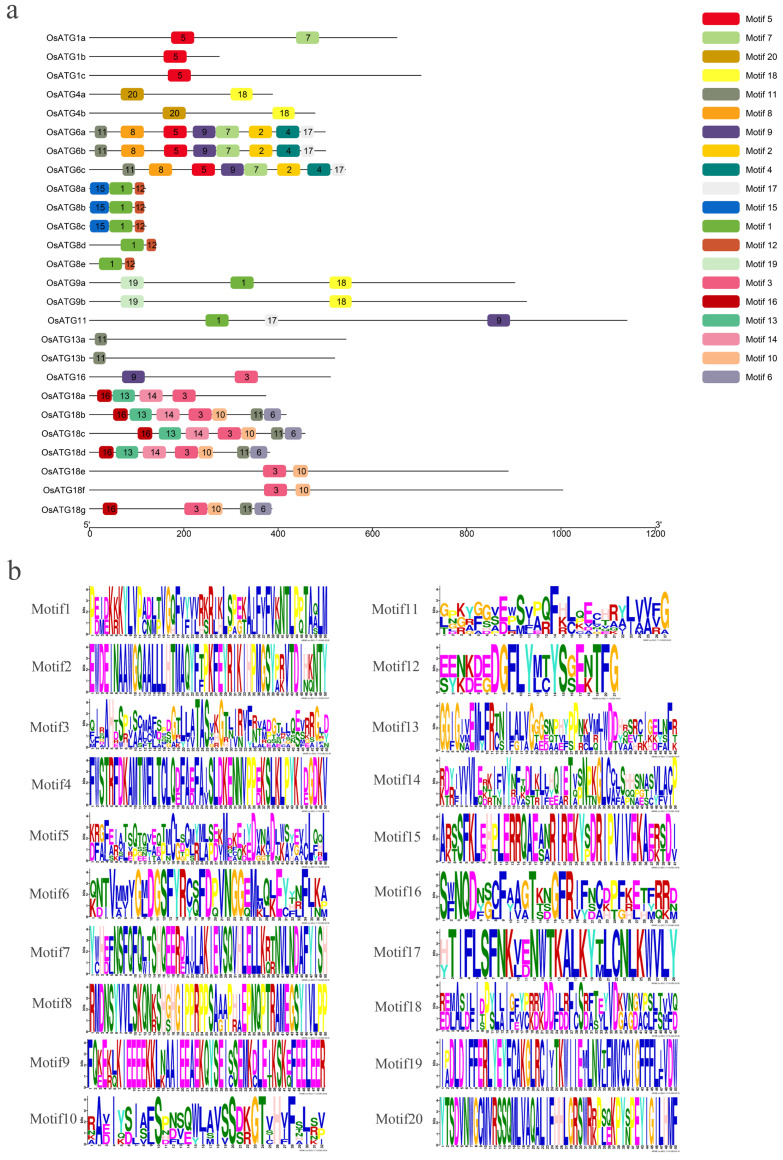
Motif analysis of OsATGs. (**a**) The distribution of 20 motifs in OsATGs. The scale represents the protein size of OsATGs, and the different colored boxes indicate the presence of different motifs. (**b**) The sequences of the 20 motifs in OsATGs. The height of each amino acid symbol is proportional to the conservation degree of the common sequence depicted by the 20 motifs. The shortest motif (motif 12) is only 21 amino acids and the longest motifs (motifs 1–5, 7, 8, 14, 19, and 20) are 50 amino acids. These sequence motifs are marked as created by MEME (motif-based sequence analysis tools).

**Figure 4 plants-13-00927-f004:**
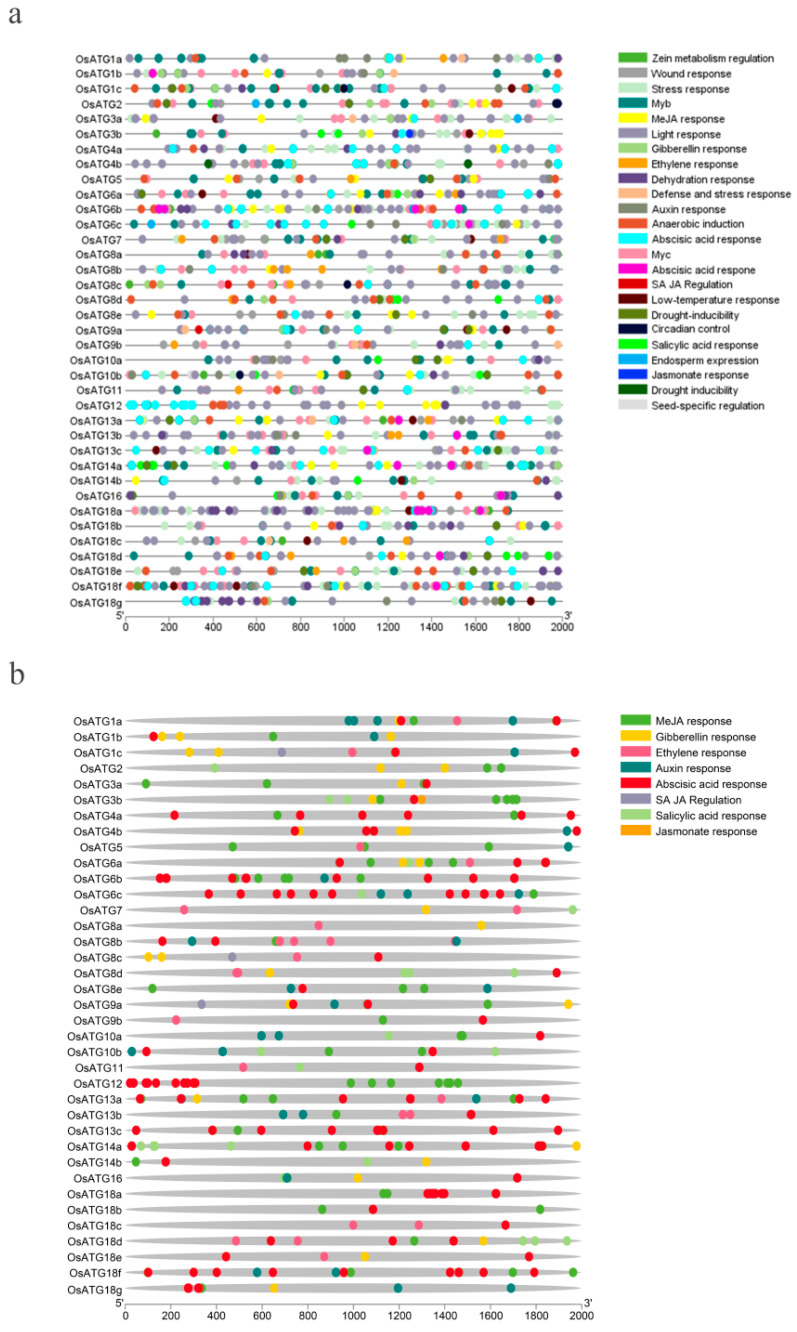
Cis-acting elements analysis of OsATGs. (**a**) The distribution of all cis-acting elements in OsATGs. The scale represents the sequence length of 2kb upstream of the *OsATG* genes, and the boxes of different colors represent different cis-acting elements (CREs). There are a total of 24 CREs distributed in the promoter regions of 37 OsATGs. (**b**) The distribution of cis-acting elements related to plant hormones in OsATGs. The scale represents the sequence length of 2kb upstream of the *OsATG* genes, and the boxes of different colors represent different cis-acting elements. There are a total of 7 CREs related to plant hormones distributed in the promoter regions of 37 OsATGs.

**Figure 5 plants-13-00927-f005:**
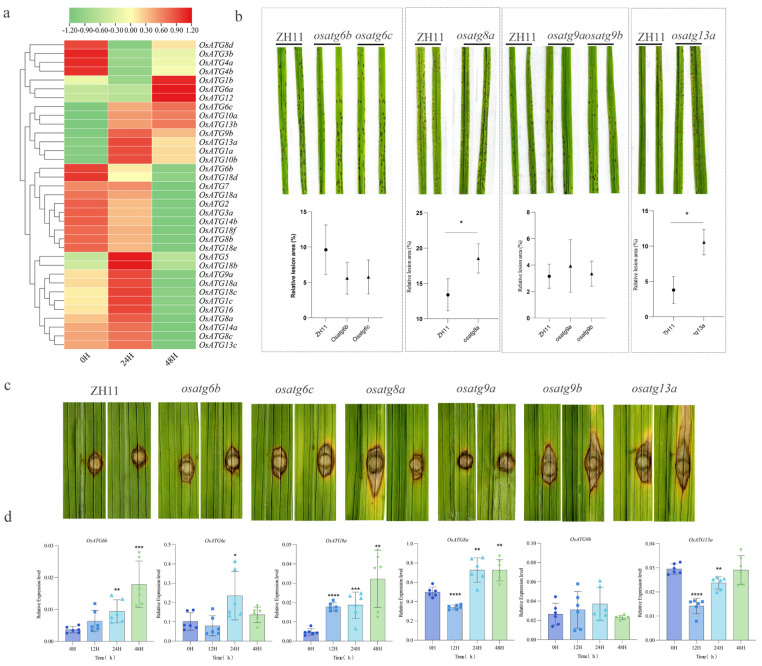
Resistance of OsATG mutants to rice blast fungus. (**a**) Analyzing the expression pattern of OsATGs during *M. oryzae* infection at 0 h, 24 h, and 48 h using transcriptome data. The color gradient from green to red visually represents the regulation gradient from downregulation (green) to upregulation (red). (**b**) Resistance evaluation of OsATG mutants using spray inoculation. The 2-week-old rice seedlings were inoculated with *M. oryzae* conidial suspension of 1 × 10^4^ mL^−1^. The asterisk (*) indicates statistical significance at *p* < 0.05 (*t*-test). (**c**) Resistance evaluation of OsATG mutants using punch inoculation. The 8-weeks-old rice leaves were punched and injected with *M. oryzae* conidial suspension of 50 × 10^4^ mL^−1^. (**d**) Expression of selected *OsATG* genes during *M. oryzae* infection using qRT-PCR. The gene expression levels were calculated by the method of 2^−ΔΔCT^, where ΔT is calculated by subtracting the cycle threshold Ct value of the target gene from the Ct value of the Internal reference gene. Statistical significance levels were indicated by asterisks (* for *p* < 0.05,** for *p* < 0.01, *** for *p* < 0.001, **** for *p* < 0.0001, *t*-test).

**Figure 6 plants-13-00927-f006:**
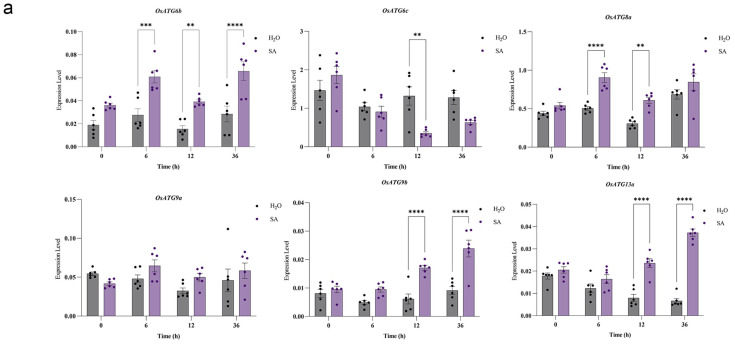
The expression of selected OsATGs in response to plant hormones. (**a**) The expression of selected OsATGs in response to SA using qRT-PCR. A concentration of 1 mmol SA was sprayed on rice seedlings of 3 weeks old. Black color indicates water treatment, while purple color represents SA treatment. (**b**) The expression of selected OsATGs in response to JA using qRT-PCR. A concentration of 0.1 mmol MeJA was sprayed on rice seedlings of 3 weeks old. Black color indicates water treatment, while purple color represents JA treatment. The target gene expression levels were calculated by the method of 2^−ΔΔCT^, where ΔT is calculated by subtracting the cycle threshold Ct value of the target gene from the Ct value of the internal reference gene. Statistical significance levels were indicated by asterisks (* for *p* < 0.05,** for *p* < 0.01, *** for *p* < 0.001, **** for *p* < 0.0001, *t*-test).

**Figure 7 plants-13-00927-f007:**
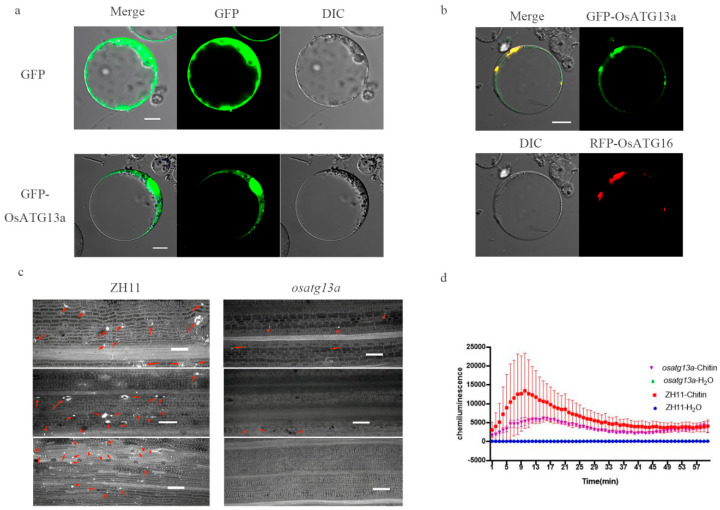
Localization of OsATG13a and PTI responses of the *osatg13a* mutant. (**a**) Localization of OsATG13a in rice protoplast. Bar represents 10 μm. GFP can be localized throughout the entire cell, and GFP-OsATG13a has strong localization in the nucleus and small spots in the cytoplasm. (**b**) Co-localization of GFP-OsATG13a and RFP-OsATG16 in rice protoplast. Bar represents 10 μm. The yellow light at Merge is composed of a combination of green and red light, indicating that GFP-OsATG13a and RFP-OsATG16 share a common subcellular localization. (**c**) Callose deposition assay of ZH11 and *osatg13a* mutant. The bright white dots indicated by the red arrows are callosum, and the bar represents 100 μm. The more protein A there is, the stronger the plant’s PTI response. (**d**) ROS burst assay of ZH11 and *osatg13a* mutant. The blue line represents water treatment in ZH11, the red line represents chitin treatment in ZH11, the green line represents water treatment in the *osatg13a* mutant, and the purple line represents chitin treatment in the *osatg13a* mutant. The higher the peak value, the stronger the plant’s PTI response.

**Figure 8 plants-13-00927-f008:**
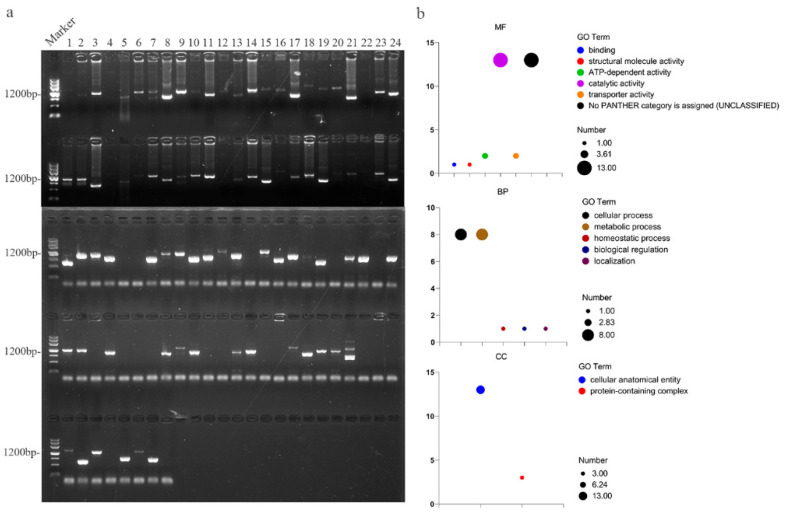
Screening and analysis of interacting proteins of OsATG13a. (**a**) PCR validation of yeast transformants. Maker represents maker III, and the sizes of the 7 bands in maker III are 4500, 3000, 2000, 1200, 800, 500, and 200 bp, respectively. The size range of bands used for sequencing is 500–2000 bp. (**b**) GO analysis of potential interacting proteins of OsATG13a. MF represents Molecular Function, CC represents Cellular Component, and BP represents Biological Process. Different colors of circles represent different categories, and the size of the circles represents the number of clusters.

**Table 1 plants-13-00927-t001:** Physicochemical properties and subcellular localization prediction of OsATGs.

Gene Name	Amino Acid	Molecular Weight	Theoretical pI	Instability Index	Aliphatic Index	GRAVY	Subcellular Localization
OsATG1a	652	72,903.85	6.21	40.47	92.33	−0.299	nucleus
OsATG1b	275	30,840.6	8.49	55.13	102.36	−0.054	nucleus
OsATG1c	703	77,457.54	6.88	56.48	84.22	−0.403	endomembrane system
OsATG2	1919	212,230.11	5.07	44.47	85.83	−0.265	endomembrane system
OsATG3a	294	33,057.95	4.52	47.56	76.22	−0.469	endomembrane system
OsATG3b	212	23,369.21	4.58	42.9	72.59	−0.478	nucleus
OsATG4a	388	42,519.74	4.64	37.54	82.22	−0.168	nucleus
OsATG4b	478	52,525.64	4.66	52.67	78.33	−0.252	nucleus
OsATG5	380	42,348.98	5	55.69	83.11	−0.377	organelle membrane
OsATG6a	500	55,898.42	5.96	39.39	79.26	−0.362	nucleus
OsATG6b	501	56,254.86	5.86	42.41	75.75	−0.418	nucleus
OsATG6c	543	61,148.31	5.95	42.94	80.29	−0.414	organelle membrane
OsATG7	1041	113,579.91	5.78	36.32	92.1	−0.110	nucleus
OsATG8a	119	13,663.71	8.78	40.57	81.09	−0.469	nucleus
OsATG8b	119	13,732.86	8.78	40.36	89.33	−0.403	nucleus
OsATG8c	120	13,888.07	8.8	38.82	88.58	−0.364	nucleus
OsATG8d	142	16,008.41	6.89	63.38	73.45	−0.297	nucleus
OsATG8e	96	10,864.61	6.89	43.08	83.12	0.015	nucleus
OsATG9a	902	103,663.59	6.15	52.01	86.65	−0.231	nucleus
OsATG9b	927	106,325.6	6.96	51.01	82.84	−0.272	nucleus
OsATG10a	198	22,303.36	5.81	42.93	81.16	−0.305	nucleus
OsATG10b	185	20,232.67	4.64	45.96	87.41	−0.185	nucleus
OsATG11	1140	126,809.03	5.77	41.66	82.09	−0.490	nucleus
OsATG12	93	10,461.01	9.05	37.9	92.26	−0.101	nucleus
OsATG13a	544	57,756.13	9.19	67.32	67.9	−0.341	nucleus
OsATG13b	520	55,863.62	8.46	70.96	65	−0.499	nucleus
OsATG13c	601	65,461.48	7.29	62.02	75.49	−0.485	nucleus
OsATG14a	387	42,764.6	8.48	59.28	87.31	−0.400	nucleus
OsATG14b	487	54,156.64	9.71	55.13	76.32	−0.514	nucleus
OsATG16	511	56,463.94	6.82	46.52	83.07	−0.396	nucleus
OsATG18a	374	40,875.54	8.87	45.23	87.38	−0.140	endomembrane system
OsATG18b	417	46,186.52	8.33	37.42	78.08	−0.196	endomembrane system
OsATG18c	457	48,721.81	6.12	49.43	75.36	−0.296	endomembrane system
OsATG18d	382	42,553.34	6.51	32.37	76.07	−0.170	plasma membrane
OsATG18e	888	95,991.19	6.57	46.55	78.5	−0.315	endomembrane system
OsATG18f	1004	108,586.38	5.67	46.9	77.19	−0.275	nucleus
OsATG18g	387	42,221.48	8	49.56	69.77	−0.311	nucleus

**Table 2 plants-13-00927-t002:** Possible interacting proteins that may participate in immunity.

Name in Study	Gene ID	Function Prediction	Reference about Immunity
OsRLK170576	XP_015639675.1	LRR-like protein kinase (RLK ID: OsatE170576)	Yin and Liu 2023 [33]
OsKEG1	XP_015639441.1	E3 ubiquitin-protein ligase KEG	Gao et al., 2021 [34]
OsANR1	XP_015634145.1	Anthocyanidin reductase ANR	Xin et al., 2020 [35]
OsPP2A-4	XP_015632306.1	Serine/threonine-protein phosphatase PP2A-4	Muñiz García et al., 2022 [36]
OsVPE1	XP_015619322.1	Vacuolar-processing enzyme VPE	Zhang et al., 2010 [37]
OsMDH1	XP_015634484.1	Malate dehydrogenase MDH	Zhou et al., 2023 [38]

## Data Availability

Data supporting the findings of this study are available within the article or its Appendix A.

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
