# Peer review of "Comprehensive Analysis of Autophagy-Related Genes in Rice Immunity against Magnaporthe oryzae"

_plants, 2024, doi:10.3390/plants13070927_

Round 1

Reviewer 1 Report

Comments and Suggestions for Authors

This article focuses on the role of autophagy in rice plant defense against Magnaporthe oryzae.

Firstly, the introduction provides a comprehensive background on the topic of autophagy, its related genes (ATGs), and it discusses the involvement of autophagy in the growth and development of plant. Also, it includes relevant references that support the information presented.

Secondly, the research design is robust, employing a combination of bioinformatic analysis, molecular biology techniques, and plant pathology assays, which are adequately described.

Finally, the conclusions drawn from these results are well supported. The study clearly demonstrates the involvement of specific OsATGs in the immune response of rice to M. oryzae; the subcellular localization and immune response analysis of OsATG13a, further validate the role of these genes. Also, the results are in line with other studies, reinforcing their conclusions.

Therefore, I would accept this article in the present form, due to I did not detect any issue or concern about the design, result and discussion of this research.

Author Response

Response to Reviewer 1 Comments

1. Summary

Dear reviewer:

Thank you very much for taking the time to review this manuscript.

2. Questions for General Evaluation

Reviewer’s Evaluation

Response and Revisions

Does the introduction provide sufficient background and include all relevant references?

Yes

Thanks for your comments

Are all the cited references relevant to the research?

Yes

Thanks for your comments

Is the research design appropriate?

Yes

Thanks for your comments

Are the methods adequately described?

Yes

Thanks for your comments

Are the results clearly presented?

Yes

Thanks for your comments

Are the conclusions supported by the results?

Yes

Thanks for your comments

3. Point-by-point response to Comments and Suggestions for Authors

Comments 1: Firstly, the introduction provides a comprehensive background on the topic of autophagy, its related genes (ATGs), and it discusses the involvement of autophagy in the growth and development of plant. Also, it includes relevant references that support the information presented.

Response 1: Thank you for your thoughtful review. 

Comments 2: Secondly, the research design is robust, employing a combination of bioinformatic analysis, molecular biology techniques, and plant pathology assays, which are adequately described.

Response 2: Thank you for your thoughtful review.

Comments 3: Finally, the conclusions drawn from these results are well supported. The study clearly demonstrates the involvement of specific OsATGs in the immune response of rice to M. oryzae; the subcellular localization and immune response analysis of OsATG13a, further validate the role of these genes. Also, the results are in line with other studies, reinforcing their conclusions.

Response 3: Thank you for your thoughtful review. 

4. Response to Comments on the Quality of English Language

Point 1: English language fine. No issues detected

Response 1: Thanks for your comments.

5. Additional clarifications

None

Reviewer 2 Report

Comments and Suggestions for Authors

The work carries out a detailed characterization of the genes associated with autophagy in rice. They find that the regulatory regions (promoters) of these genes contain cis-actin-elements that respond to salicylate and jasmonate, associated with plant defense.

In loss-of-function experiments they demonstrate that these genes contribute to resistance against rice blight. In particular, the genes OsATG8a and OsATG13a seem to be the genes with the greatest participation in resistance to M.oryzae.

In response to inoculation with M.oryzae, the expression of OsATG8a and 13a increases mainly after 12 hours. However, OsATG13a has a very high expression at time 0 hrs. It would be convenient to comment on what this means in relation to its participation in tolerance to the disease.

OsATG8a and Osatg13a respond to SA by inducing their expression at all times analyzed. The difference is that OsATG13b has an expression between 2 and 3 times higher than in plants not treated with the hormone. Similarly, in response to JA the expression of OsATG13a is much higher compared to plants without hormone. I consider again that these differences and their relationship with tolerance to M. oryzae be commented on.

I suggest commenting on whether the tolerance levels are related to the expression levels of the OsATG's genes after inoculation with M.oryzae and if this is related to the levels of JA and AS. This would give a more defined view of the function of these genes (OsATG8a and OsATG13a) in tolerance mechanisms.

Finalmente, la identificación de proteínas asociadas a la respuesta inmune que interaccionan con OsATG13a, abre un camino que permitirá la identificación de nuevas proteínas responsables de la tolerancia a M.oryzae.

Finally, the identification of proteins associated with the immune response that interact with OsATG13a opens a path that will allow the identification of new proteins responsible for tolerance to M.oryzae.

Author Response

Response to Reviewer 2 Comments

1. Summary

Dear reviewer:

Thank you very much for taking the time to review this manuscript. These comments have been very helpful to us, and our answers can be found in the following text.

2. Questions for General Evaluation

Reviewer’s Evaluation

Response and Revisions

Does the introduction provide sufficient background and include all relevant references?

Yes

Thanks for your comments

Are all the cited references relevant to the research?

Yes

Thanks for your comments

Is the research design appropriate?

Yes

Thanks for your comments

Are the methods adequately described?

Yes

Thanks for your comments

Are the results clearly presented?

Yes

Thanks for your comments

Are the conclusions supported by the results?

Can be improved

Thanks for your comments

3. Point-by-point response to Comments and Suggestions for Authors

Comments 1: In response to inoculation with M.oryzae, the expression of OsATG8a and 13a increases mainly after 12 hours. However, OsATG13a has a very high expression at time 0 hrs. It would be convenient to comment on what this means in relation to its participation in tolerance to the disease.

Response 1: Thank you for your question. It is challenging to explain why the expression of OsATG8a and 13a at 12h was lower than at 0h. This could be due to inhibitory factors that may have suppressed the expression of these two genes initially. Subsequently, at 24h and 48h, it appears that these inhibitory factors were alleviated, leading to an increase in gene expression. So, their participations in tolerance are not reflected in this process. Future research could focus on identifying and understanding the specific factors that inhibit the expression of these genes during the early stages of infection.

Comments 2: OsATG8a and Osatg13a respond to SA by inducing their expression at all times analyzed. The difference is that OsATG13b has an expression between 2 and 3 times higher than in plants not treated with the hormone. Similarly, in response to JA the expression of OsATG13a is much higher compared to plants without hormone. I consider again that these differences and their relationship with tolerance to M. oryzae be commented on.

Response 2: Thank you for your suggestion. SA and JA have been established as key hormones in mediating rice resistance to M. oryzae. These hormones are synthesized abundantly within rice plants following M. oryzae infection. Treatment of rice with JA and SA has been shown to stimulate the expression of a series of defense-related genes. In our research, both SA and JA treatments led to an upregulation in the expression of OsATG8a and Osatg13a, suggesting their involvement in the resistance against M. oryzae. The varying levels of expression of these genes may indicate differing degrees of their contribution to the defense mechanism.

Comments 3: I suggest commenting on whether the tolerance levels are related to the expression levels of the OsATG's genes after inoculation with M.oryzae and if this is related to the levels of JA and AS. This would give a more defined view of the function of these genes (OsATG8a and OsATG13a) in tolerance mechanisms.

Response 3: Thank you for your suggestion. If the expression of OsATGs is upregulated after M. oryzae inoculation, it strongly suggests their involvement in resistance against M. oryzae. Additionally, it is believed that the levels of JA and SA in rice increase following M. oryzae inoculation. However, further experimental events are required to definitively determine the function of these genes in resistance. In Figure 5C of our paper, we present the significant decrease in resistance to M. oryzae observed in mutants of OsATG8a and OsATG13a.

4. Response to Comments on the Quality of English Language

Point 1: I am not qualified to assess the quality of English in this paper

Response 1: Thanks for your comments.

5. Additional clarifications

None

Reviewer 3 Report

Comments and Suggestions for Authors

Dear Editor-in-chief,

This manuscript entitled Comprehensive Analysis of Autophagy-Related Genes in Rice 2 Immunity against Magnaporthe oryzae delivers interesting data and excellent interpretation. Therefore, this manuscript can be acceptable to Plants; however, there are small minor revisions required for the final publication. Due to the quantity and quality of these experiments, this manuscript will be an excellent paper published in plants. The minor revision points are as follow:

Minor points:

1.     What is the difference between ATG and OsATG? The authors may add a sentence describing the difference in introduction section.

2.     Line 97: Please remove the underline from Oryza sativa and ATGs.

3.     Line 195: What’s the meaning of MEME? Please add its full form.

4.     Line 218: Write as M. oryzae” in rice

5.     Line 290 and 391: Please use only ROS

6.     Line 298: Use a space  before co-colonization

7.     Line 332: O. sativa should be in italic

8.     Line 336: Remove the extra dot at the end the sentence

9.      Line 364 and 365: Check “a concentration of 105 mL-1”. Is it correct?

10.  Line 405-406: Revise the sentence

11.  Line 467: M. oryzae should be in italic

12.  Line 471:  Blumeria graminis f.sp. tritici

13.  Line 500: Please write N. benthamiana instead of Nicotiana benthamiana

14.  Line 504: Xanthomonas manihotis should be in italic

This manuscript was very well written based on many excellent experiments; therefore, there will be no point of further recirculation of the revised version of this manuscript

In addition, this manuscript does not need any English editing.  

Author Response

Response to Reviewer 3 Comments

1. Summary

Dear reviewer:

Thank you very much for taking the time to review this manuscript. We add a sentence describing the difference between ATG and OsATG in introduction section. And we have revised all minor revision points.

2. Questions for General Evaluation

Reviewer’s Evaluation

Response and Revisions

Does the introduction provide sufficient background and include all relevant references?

Yes

Thanks for your comments

Are all the cited references relevant to the research?

Yes

Thanks for your comments

Is the research design appropriate?

Yes

Thanks for your comments

Are the methods adequately described?

Yes

Thanks for your comments

Are the results clearly presented?

Yes

Thanks for your comments

Are the conclusions supported by the results?

Yes

Thanks for your comments

3. Point-by-point response to Comments and Suggestions for Authors

Comments 1: What is the difference between ATG and OsATG? The authors may add a sentence describing the difference in introduction section.

Response 1: Thank you for pointing this out. We agree with this comment. Therefore, We add a sentence describing the difference between ATG and OsATG in introduction section. This change can be found in line 93.

Comments 2: Line 97: Please remove the underline from Oryza sativa and ATGs; Line 195: What’s the meaning of MEME? Please add its full form; Line 218: Write as “M. oryzae” in rice; Line 290 and 391: Please use only ROS; Line 298: Use a space  before co-colonization; Line 332: O. sativa should be in italic; Line 336: Remove the extra dot at the end the sentence; Line 364 and 365: Check “a concentration of 105 mL-1”. Is it correct?; Line 405-406: Revise the sentence; Line 467: M. oryzae should be in italic; Line 471: Blumeria graminis f.sp. tritici; Line 500: Please write N. benthamiana instead of Nicotiana benthamiana; Line 504: Xanthomonas manihotis should be in italic.

Response 2: Thank you for pointing these out. We have revised all minor revision points. These change can be found in line 102, 203, 226, 302, 310, 345, 349, 378, 417, 480, 490, 519 and 524.

4. Response to Comments on the Quality of English Language

Point 1: English language fine. No issues detected

Response 1: Thanks for your comments

5. Additional clarifications

None
